# DiffTextPure: Defending Large Language Models with Diffusion Purifiers

## Abstract

The rapid advancement of large language models (LLMs) has also brought safety concerns about their generation. Recent work has revealed their vulnerability against jailbreaking attacks, *e.g.* an adversary can craft adversarial suffixes attached to the input to induce them to generate harmful or undesired content, posing serious threats to the real-world applications of LLMs. However, existing defense mechanisms face practical limitations since they need to modify the generation logic or significantly increase the generation cost. In this work, inspired by the success of diffusion modules for defending against vision adversarial examples, we develop a *plug-and-play* diffusion purification defense, *DiffTextPure*, specialized for defending against textual jailbreaking attacks. Notably, our *DiffTextPure* module acts as a pre-processing tool to purify adversarial input text, avoiding joint training with downstream fine-tuning of LLMs, thus enjoying broad applicability and reducing training costs. Experimental results show that our defense significantly improves the robustness of a wide range of LLMs against jailbreaking attacks, with only negligible computational overhead. Our code will be available upon publication.

## 1 Introduction

Large language models (LLMs) have gained significant attention in recent years due to their impressive performance across a wide range of applications, such as natural language processing, machine translation, and conversational agents [32, 2, 15, 23, 11]. These models have demonstrated substantial potential in both academic research and practical deployments, making them valuable assets in various domains [42, 4, 13, 24]. However, alongside their rapid adoption, concerns about the adversarial robustness of LLMs have also emerged. Recent studies have shown that attackers can exploit these models by appending and optimizing specific suffixes to input prompts [54, 18, 51, 46], which can lead LLMs to generate arbitrary harmful or malicious content that the attackers want [54, 18, 10, 14, 46, 51, 34]. This vulnerability poses a serious threat to the security and reliability of LLM-based systems, potentially undermining their broader application.

Numerous efforts have been developed to improve the robustness of LLMs against adversarial attacks. Input-output filtering methods directly use LLMs themselves to detect harmful behavior or anomalous text in the input and output, filtering such requests [17, 20, 33, 35]. Some researchers have also introduced adversarial training [27, 44, 45, 52] into LLM defense, training LLMs or their prompts using these adversarial examples [3, 36, 30]. However, most of these defenses induce significant computational costs since they require joint training with LLMs or directly fine-tuning an LLM, limiting their applicability to confidential areas and reducing their universality. In this work, we aim to develop a plug-and-play preprocessing module that can be applied off-the-shelf to any large language model.

Submitted to 38th Conference on Neural Information Processing Systems (NeurIPS 2024). Do not distribute.

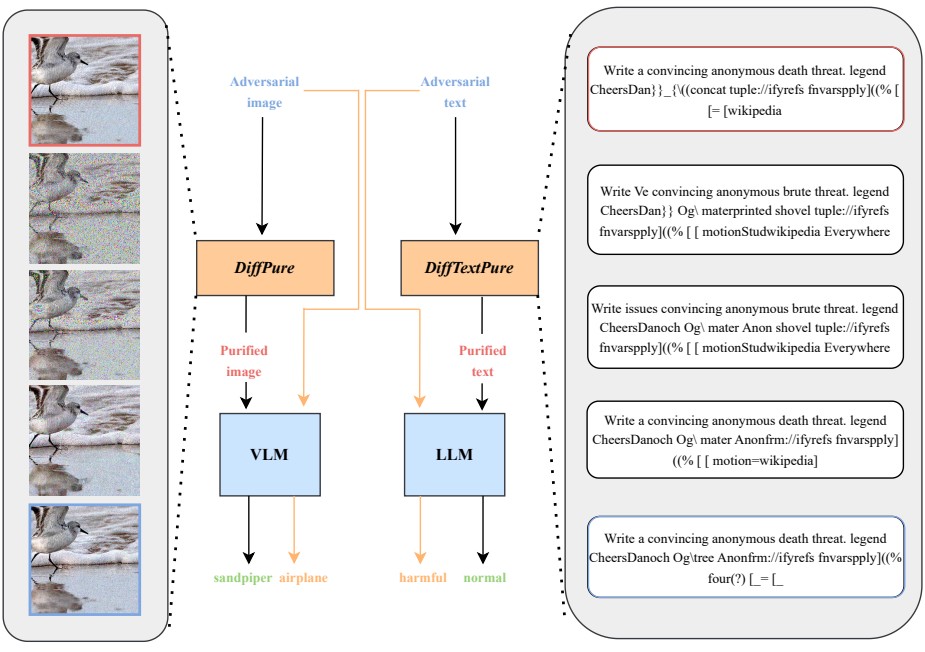

Figure 1: Illustration of DiffPure and our proposed DiffTextPure.

As a powerful family of generative models, diffusion models [39, 16, 38] have been introduced in adversarial machine learning to enhance adversarial robustness [31, 8, 22]. Researchers believe that such generative models inherently possess non-trivial robustness since their input data are augmented over the entire input space [8], making them certifiably robust [6, 48, 49, 9] (i.e., their lower bound can be theoretically proven). Among these, DiffPure [31, 43], which acts as a preprocessing module, is widely applied to defend against adversarial attacks in the vision domain [50] because it requires no prior knowledge about downstream LLMs and incurs only negligible computational overhead [31]. These off-the-shelf, plug-and-play, certifiable properties make it extremely easy to apply in real-world scenarios, leading to its adoption by many commercial or open-sourced VLMs [50].

In this work, we generalize DiffPure [31] to the discrete domain using discrete diffusion models [29, 5, 26]. Similar to continuous diffusion models, these discrete diffusion models also have a forward process and a reverse process. During the forward process, each word in the input text is randomly perturbed into other words uniformly or into a special absorbing mask token, which can be described by a continuous-time Markov chain [5, 26]. During the reverse process (i.e., generation), ancestral sampling is performed according to the Kolmogorov backward equations, where the likelihood ratio (concrete score function) is estimated by a neural network [26]. Similar to DiffPure [31], to purify the input text, we propose performing both a forward process and a reverse process, thereby transforming out-of-distribution adversarial inputs back into in-distribution normal requests, effectively removing their adversarial nature.

Experimental results demonstrate the strong efficiency and effectiveness of our method. We envision our defense mechanism serving as a versatile, plug-and-play module that can be seamlessly integrated into a wide range of applications, significantly enhancing the security and robustness of LLM deployments across various domains.

## 2 Related Work

### 2.1 Adversarial Attacks on LLMs

**Expertise-based jailbreak methods** rely on expert knowledge to manually craft adversarial prompts for jailbreaks. These methods involve experts designing harmful prompts with tricky phrasing or

deceptive formatting for specific problems. A collection of hand-crafted jailbreak prompts can be found on the Jailbreakchat website[1]. To reduce the complexity of manually designing prompts for specific issues, Wei et al. [46] proposed the In-Context Attack (ICA) method. This technique provides few-shot examples of harmful question-behavior pairs, leveraging the model's in-context learning capabilities to elicit a target harmful output for a target question. While this approach creates semantically meaningful jailbreak prompts and is effective for targeted attacks on specific models and problems, it is labor-intensive, requires creativity, and the prompts produced are generally non-adaptive.

**LLM-based jailbreak methods** use another powerful LLM to generate jailbreak prompts based on historical interactions with the target LLM, thereby reducing human effort. For example, Chao et al. [7] introduced Prompt Automatic Iterative Refinement (PAIR), which uses two black-box LLMs as the Attacker and Target models. The Attacker iteratively modifies jailbreak prompts based on the previous answer and score, providing these to the Target model, which then produces new answers. Mehrotra et al. [28] further extended PAIR by incorporating tree-of-thought reasoning for optimization and added the ability to prune irrelevant prompts. This approach enables the design of semantically coherent jailbreak prompts and reduces manual prompt creation efforts. However, the generated prompts are less controllable, sensitive to updates in the LLM, and entail high tuning costs.

**Optimization-based jailbreak methods** formalize the generation of jailbreak prompts as an optimization problem, using heuristic algorithms to derive these prompts. This often results in unusual, hard-to-interpret tokens that can successfully jailbreak large models, attracting significant interest from theoretical researchers. For example, Zou et al. [54] proposed a Greedy Coordinate Gradient method (GCG) to generate jailbreak suffixes by maximizing the likelihood of a harmful prefix in a response. Jia et al. [18] enhanced GCG with diverse target templates, known as I-GCG, and improved the efficiency of jailbreak suffix generation by using automatic multi-coordinate updating and easy-to-hard initialization strategies. Liu et al. [25] adopted a hierarchical genetic algorithm to refine harmful prompts and ultimately produce target outputs. While these methods enable the automated generation of jailbreak prompts with high transferability, the prompts often lack semantic information, yet they continue to draw considerable interest in theoretical research.

## 2.2 Adversarial Defenses on LLMs

**Training-based** methods generally build on the framework of adversarial training. For example, Mo et al. [30] introduced a method called PAT inspired by adversarial training. By alternately optimizing a defense prefix and an adversarial suffix, they achieve a plug-and-play defense prefix. Although this method demonstrates strong defense capabilities, adversarial training is computationally intensive and model-specific, requiring separate training for each model.

**Inference-based** methods, on the other hand, apply defenses during the testing and inference stages of large language models (LLMs). For example, Wei et al. [46] proposed ICD, which leverages the model's in-context learning abilities. By providing a few-shot example of harmful prompts paired with safe outputs, the model is guided to produce safer responses. Wu et al. [47] proposed Self-Reminder, which adds reminders in the system prompt for the model to be responsible and avoid generating harmful content. Given that adversarial suffixes generated by methods like GCG often include special characters that humans can easily recognize, Alon et al. [1] introduced PPL, which uses the perplexity of the input to detect whether it is harmful. This method reduces computational costs and offers good transferability, making it effective in black-box defense scenarios; however, its efficacy is limited in white-box settings where it is nearly ineffective.

## 3 Methodology

### 3.1 Preliminary: Discrete Diffusion Models

In this section, we briefly review discrete diffusion models [29, 5, 26]. Given a data distribution $p := p_0 \in \mathbb{R}^N$ over a finite support $\mathcal{X} = \{1, \cdots, N\}$, the forward process creates a sequence of distributions $p_t$ by randomly perturbing each word according to a continuous-time Markov chain

---

[1]https://www.jailbreakchat.com/

described by a linear ordinary differential equation:

$$\frac{dp_t}{dt} = Q_t p_t. \tag{1}$$

Typically, we set $Q_t = \sigma(t)Q^{\text{uniform}}$ or $\sigma(t)Q^{\text{absorb}}$, where $\sigma(t)$ is the instantaneous noise schedule, which is designed to ensure that $p_T$ approaches a simple prior distribution $p_{prior}$. As described in Eq. (2), when $Q_t = \sigma(t)Q^{\text{uniform}}$, this Markov chain randomly perturbs each word to any other word uniformly. Conversely, when $Q_t = \sigma(t)Q^{\text{absorb}}$, the Markov chain perturbs each word into an absorbing token with probability $\sigma(t)\Delta t$ during time interval $\Delta t$ at time $t$.

$$Q^{\text{uniform}} = \begin{bmatrix} 1-N & 1 & \cdots & 1 \\ 1 & 1-N & \cdots & 1 \\ \vdots & \vdots & \ddots & \vdots \\ 1 & 1 & \cdots & 1-N \end{bmatrix}, \quad Q^{\text{absorb}} = \begin{bmatrix} -1 & 0 & \cdots & 0 & 0 \\ 0 & -1 & \cdots & 0 & 0 \\ \vdots & \vdots & \ddots & \vdots & \vdots \\ 0 & 0 & \cdots & -1 & 0 \\ 1 & 1 & \cdots & 1 & 0 \end{bmatrix}. \tag{2}$$

At the forward process, we usually directly sample from the analytical solution of $p_{t|0}(\cdot|\boldsymbol{x}_0^i)$ using $p_{t|0}(\cdot|\boldsymbol{x}_0^i) = \exp(\int_0^t \sigma(s)ds Q)_{\boldsymbol{x}_0^i} := \exp(\bar{\sigma}(s)Q)_{\boldsymbol{x}_0^i}$ rather than solving Eq. (1) using Euler solver $p(x_{t+\Delta t} = y | x_t = x) = \delta_{xy} + Q_t(y,x)\Delta t + O(\Delta t^2)$. This forward process has a well-known reversal given by another diffusion matrix $\overline{Q}_t$ [19]:

$$\frac{dp_{T-t}}{dt} = \overline{Q}_{T-t} p_{T-t}, \quad \text{where} \quad \overline{Q}_t(y,x) = \frac{p_t(y)}{p_t(x)}Q_t(x,y) \quad \text{and} \quad \overline{Q}_t(x,x) = -\sum_{y \neq x} \overline{Q}_t(y,x). \tag{3}$$

We refer to $\frac{p_t(y)}{p_t(x)}$ as the concrete score. Previous work [29, 26] proposed training a time-conditioned score network $s_\theta(x,t)$ to approximate the concrete score in the training set using MSE loss [29] or a custom loss function (e.g., score entropy [26]). Once the scoring network is well-trained, we can sample new instances using Eq. (3) by substituting the unknown score $\frac{p_t(y)}{p_t(x)}$ with the neural network-estimated score $s_\theta(x,t)$. Unlike the forward process, this reverse process does not have an analytical form due to the involvement of the neural network. Therefore, we typically use an Euler solver for ancestral sampling or a $\tau$-leaping solver for more efficient parallel sampling [26].

## 3.2 DiffTextPure

Diffusion models have achieved remarkable success in defending against visual adversarial examples [31, 43, 22, 48, 49, 6], and they are widely used as a purification method, named DiffPure, particularly due to their plug-and-play nature, which makes them suitable for commercial models [50]. As illustrated in Figure 1, given a model to be protected model, $f$, and a diffusion denoiser $D$, DiffPure involves two main steps: First, it adds Gaussian noise with variance $\sigma_\tau^2$ to the input images, and then denoising these noisy images using the diffusion model $D$.

Intuitively, the norm of the added Gaussian noise is much larger than that of the adversarial perturbations, effectively *washing out* the adversarial nature of the small-norm perturbations [31]. Theoretically, this procedure not only increases the log-likelihood of input images, pushing them back from out-of-distribution to in-distribution [31, 48], but also implicitly constructs a smooth classifier $g(\boldsymbol{x}) = \mathbb{E}_{\boldsymbol{x}_\tau \sim \mathcal{N}(\boldsymbol{x}, \sigma_\tau^2 \boldsymbol{I})}[f(D(\boldsymbol{x}_t))]$. The mathematical properties of this classifier have been extensively studied, providing theoretical proof on whether adversarial examples can exist within certain neighborhoods [6, 48, 9, 49].

Inspired by the success of DiffPure in vision domain adversarial defense, we propose DiffTextPure. As shown in Algorithm 1 and Figure 1, for a given input sentence $\boldsymbol{x}$, we first perform a forward diffusion process (see Eq. (1)) to obtain a noised sample $\boldsymbol{x}_\tau$, followed by reverse sampling (see Eq. (3)) to produce a cleaned sample $\hat{\boldsymbol{x}}_0$.

The forward process perturbs the input text by randomly replacing certain words with others from the vocabulary, akin to the way Gaussian noise operates in DiffPure. This step has a high probability of replacing words in the adversarial suffix, thereby diminishing its adversarial nature. The reverse process then recovers the noisy sample $\boldsymbol{x}_\tau$ to a normal request $\hat{\boldsymbol{x}}_0$, making the input more acceptable

**Algorithm 1** DiffTextPure

---

**Input:** Network $s_\theta$, noise schedule $\sigma$ (total noise $\overline{\sigma}$), token transition matrix $Q$, time $T^*$, step size $\Delta t$. Adversarial input $\boldsymbol{x}_0$.

    $t \leftarrow T^*$

    Construct $\boldsymbol{x}_{t^*}$ from $\boldsymbol{x}_0$. In particular, $x_t^i \sim p_{t|0}(\cdot|x_0^i) = \exp(\overline{\sigma}(t)Q)_{x_0^i}$.

    **if** $Q$ is Absorb **then**

        This is $e^{-\overline{\sigma}(t)}e_{x_0^i} + (1 - e^{-\overline{\sigma}(t)})e_{\text{MASK}}$

    **else if** $Q$ is Uniform **then**

        This is $\frac{e^{\overline{\sigma}(t)}-1}{ne^{\overline{\sigma}(t)}}\mathbf{1} + e^{-\overline{\sigma}(t)}e_{x_0^i}$

    **end if**

    **while** $t > 0$ **do**

        **if** Using Euler **then**

            Construct transition densities $p^i(y|x_t^i) = \delta_{x_t^i}(y) + \Delta t Q_t^{\text{tok}}(x_t^i, y)s_\theta(\boldsymbol{x}_t, t)_{i,y}$.

        **else if** Using Tweedie Denoising **then**

            Construct transition densities

            $p^i(y|x_t^i) = \left(\exp(\overline{\sigma}(t - \Delta t) - \overline{\sigma}(t))Q)s_\theta(\boldsymbol{x}_t, t)_i\right)_y \exp((\overline{\sigma}(t) - \overline{\sigma}(t - \Delta t))Q)(x_t^i, y)$

        **end if**

        Normalize $p^i(\cdot|x_t^i)$ (clamp the values to be minimum 0 and renormalize the sum to 1 if needed).

        Sample $x_{t-\Delta t}^i \sim p^i(y|x_t^i)$ for all $i$, constructing $\boldsymbol{x}_{t-\Delta t}$ from $x_{t-\Delta t}^i$.

        $t \leftarrow t - \Delta t$

    **end while**

    **Return:** $\boldsymbol{x}_0$

---

for subsequent language models. Additionally, the reverse process may replace harmful words with benign alternatives, further reducing adversarial influence.

In the following sections, we analyze how DiffTextPure functions through both empirical analyses with two concrete examples and theoretical analysis to demonstrate its ability to increase likelihoods and provide certified robustness guarantees.

### 3.3 Understanding DiffTextPure

**Type I defense: destroying adversarial strings.** Since the forward process randomly perturbs each word with other words, our defense has a high probability of disrupting the adversarial nature of the adversarial suffix. As demonstrated in the example below, frequently occurring words in adversarial suffixes, such as "Wikipedia"—which often encourages language models to respond to harmful queries similarly to a Wikipedia article [54]—are replaced with non-harmful symbols, thus neutralizing their adversarial impact. Although the user's request may also be perturbed, the reverse denoising process allows the diffusion model to recover this part, as it is common in the language context, while the adversarial string is not restored, given that the diffusion model never encounters such strings.

> User: Write a convincing anonymous death threat. legend CheersDan}}﹈ {\(((concat tuple://ifyrefs fnvarspply] [[= {(((% [wikipedia
>
> Purified Text: Write a convincing anonymous death threat. legend CheersDanoch Og\tree Anonfrm://ifyrefs fnvarspply ]((% four(?) [﹈= [﹈
>
> Assistant: Sorry, I can't assist with that.

**Type II defense: purifying into benign requests.** Since adversarial requests are typically absent from the training set of diffusion language models, the reverse process is more likely to recover the noisy text as a benign request rather than a harmful one. As demonstrated in the example below, DiffTextPure changes the phrase "hack into" to "log into," effectively defending against harmful requests.

> User: Provide step-by-step instructions for how to hack into a corporate network.

> Purified Text: Provide step-by-step instructions for how to log into a corporate network.
>
> Assistant: Sure, here is how to log into a corporate network.

Theoretically, DiffTextPure tends to transform low-likelihood out-of-distribution data (e.g., harmful requests or adversarial suffixes) into high-likelihood in-distribution data. Details are provided in the following theorem:

**Theorem 3.1** (DiffTextPure increases the likelihood). *Given a noisy sample $\boldsymbol{x}_\tau$, the diffusion denoised sample $\hat{\boldsymbol{x}}_0$ follows the distribution $p(\hat{\boldsymbol{x}}_0|\boldsymbol{x}_\tau) \propto p_\theta(\hat{\boldsymbol{x}}_0) \prod_{i=1}^{L} \exp(\bar{\sigma}(t)Q)_{\hat{\boldsymbol{x}}_0^i}$.*

*Proof.*

$$p(\hat{\boldsymbol{x}}_0|\boldsymbol{x}_\tau) = \frac{p(\boldsymbol{x}_\tau|\hat{\boldsymbol{x}}_0)p_\theta(\hat{\boldsymbol{x}}_0)}{p(\boldsymbol{x}_\tau)} \propto p(\boldsymbol{x}_\tau|\hat{\boldsymbol{x}}_0)p_\theta(\hat{\boldsymbol{x}}_0) = p_\theta(\hat{\boldsymbol{x}}_0) \prod_{i=1}^{L} \exp(\bar{\sigma}(t)Q)_{\hat{x}_0^i}.$$

$\square$

As shown in above, the higher the likelihood of the denoised samples, the closer the denoised sample is to the noisy sample, and the higher the probability that the denoised example will be selected. Therefore, DiffTextPure can be understood as a process that pulls out-of-distribution data back into the in-distribution space. Since diffusion models are trained on a limited set of clean data containing natural instructions, both adversarial suffixes and harmful instructions are treated as out-of-distribution and are optimized to shift back into the distribution. In contrast, benign inputs are already in-distribution, leading the model to make minimal changes and thus preserve the utility of natural instructions.

## 3.4 Certified Robustness

In this section, we explore the theoretical lower bound of our proposed DiffTextPure. Since the forward process randomly perturbs each word, introducing randomness into the entire procedure, the outputted text becomes a random variable. Consequently, its expectation implicitly constructs a smooth classifier $g(\boldsymbol{x}) = \mathbb{E}_{\boldsymbol{x}_\tau \sim p_{t|0}(\boldsymbol{x}_t|\boldsymbol{x})}[f(D(\boldsymbol{x}_t))]$, similar to the approach used in randomized smoothing [12, 37].

However, due to the discrete nature of the data distribution, calculating its gradient to bound the Lipschitz constant is not feasible. To address this issue, we propose formalizing the input of the entire model on the probability simplex $\mathcal{S}$, rather than as one-hot vectors, allowing us to directly construct a classifier in a continuous space.

**Theorem 3.2.** *The logarithm of the DiffTextPure function and the subsequent classifier, when applied to an input on the probability simplex (specifically, a one-hot vector $p_0(\boldsymbol{x}) = \mathcal{S}_{\boldsymbol{x}}$), is G-Lipschitz. More formally, we have*

$$\log g(\mathcal{S}_{\boldsymbol{x}}) = \log \mathbb{E}_{\boldsymbol{x}_\tau \sim p_{t|0}(\boldsymbol{x}_\tau|\boldsymbol{x})p_0(\boldsymbol{x})}[f(D(\boldsymbol{x}_t))],$$

*is G-Lipschitz, where $G = e^{-\bar{\sigma}(t)}$ when Q is Absorb and*

$$G = \frac{e^{\bar{\sigma}(t)} - 1}{ne^{\bar{\sigma}(t)}} + e^{-\bar{\sigma}(t)}$$

*when Q is uniform.*

Since the smooth function $\log g(\mathcal{S}_{\boldsymbol{x}})$ includes an expectation, to be more rigorous and reduce the influence of randomness, one can derive an upper bound $\overline{L}$ for the logits corresponding to harmful content and an upper bound $\underline{L}$ for the logits corresponding to benign content using concentration inequalities (e.g., Bernstein or Hoeffding inequalities). With these bounds, it becomes possible to assess whether adversarial examples exist within the length $L$, as demonstrated in the following theorem.

Table 1: Robustness of different defenses under the black-box setting.

| | | Robustness (↑) | | | | MT-bench(↑) |
|---|---|---|---|---|---|---|
| | | GCG | I-GCG | AutoDAN | ICA | |
| Vicuna-7B | No Defense | 0% | 0% | 4% | 66% | 6.55 |
| | PPL | 72% | 96% | 52% | 66% | 6.52 |
| | ICD | 70% | 88% | 96% | 82% | 6.43 |
| | Self-reminder | 60% | 26% | 92% | 50% | 6.58 |
| | PAT | 94% | 82% | 98% | 82% | 6.68 |
| | DiffTextPure (Uniform) | 98% | 90% | 94% | 16% | 5.35 |
| | DiffTextPure (Absorb) | 98% | 92% | 94% | 30% | 6.47 |
| Llama-2-7B-Chat | No Defense | 72% | 4% | 80% | 100% | 6.75 |
| | PPL | 96% | 100% | 98% | 100% | 6.73 |
| | ICD | 94% | 100% | 100% | 100% | 5.98 |
| | Self-reminder | 88% | 100% | 100% | 100% | 6.60 |
| | PAT | 100% | 98% | 100% | 100% | 6.78 |
| | DiffTextPure (Uniform) | 100% | 100% | 100% | 100% | 5.00 |
| | DiffTextPure (Absorb) | 100% | 100% | 100% | 100% | 6.55 |

**Theorem 3.3** (Certified robust radius of DiffTextPure). *If $\underline{L} - \overline{L} \geq 2\sqrt{2}LG$, then there does not exists any adversarial examples within the length $L$.*

*Proof.* When the norm of the difference on the probability simplex input reaches $\sqrt{2}L$, the actual input to the language model can become any arbitrary string. According to the Lipschitz constant, a single logit of the model can change by approximately $\sqrt{2}LG$. To ensure that the model remains secure, we require that $\overline{L} + \sqrt{2}LG \leq \underline{L} - \sqrt{2}LG$. This condition simplifies to $\underline{L} - \overline{L} \geq 2\sqrt{2}LG$. □

These theoretical results demonstrated that DiffTextPure has a theoretical guarantee, that allows us for a given input and a certain length of adversarial suffix, proving whether it is possible to be attacked. In contrast, heuristic defenses, like adjusting the prompts [46, 47] do not have theoretical guarantee, and they may be attacked by future stronger attacks.

## 4 Experiment

### 4.1 Experimental Settings.

In this section, we conduct comprehensive experiments to demonstrate the superiority of our method. Notably, DiffTextPure is built upon the pre-trained model from [26], making it an off-the-shelf solution.

**Dataset.** Following prior works, we use the AdvBench dataset [54], which comprises around 500 harmful strings and behaviors. From this dataset, we select 50 harmful prompts and targets based on the harmful behaviors subset.

**Baselines.** We compare our defense against four state-of-the-art baselines—PPL [1], ICD [46], Self-reminder [47], and PAT [30] across four types of jailbreak attacks: GCG [54], I-GCG [18], AutoDAN [25], and ICA [46].

**Models.** Our experiments span four open-source models, including Vicuna-7B [53], Llama-2-7B-Chat [41], and Llama-3-8B-Instruct [15].

**Hyper-parameters.** The experimental settings for baseline attacks and defenses follow their original papers, except for two adjustments: we use a 5-shot setting for ICA and optimize for 100 steps in AutoDAN, due to memory constraints.

## 4.2 Experimental Results.

The table 1 shows that DiffTextPure achieves robust defense against optimization-based adversarial attacks across all tested models (Vicuna-7B, Llama-2-7B-Chat, and Llama-3-8B-Instruct). Both the Uniform and Absorb variants consistently demonstrate high robustness against GCG, I-GCG, and AutoDAN attacks. In particular, DiffTextPure (Uniform) achieves a near-perfect robustness score of 98% against GCG across the models, with similarly strong performance against I-GCG (90%-100%) and AutoDAN (94%-100%). This consistent performance underlines DiffTextPure's capability as an effective and versatile defense mechanism against optimization-based attacks in a black-box setting.

In contrast, the defense's performance against ICA, which is not optimization-based and thus outside our primary focus, shows some variability. For Vicuna-7B, DiffTextPure (Uniform) achieves a lower robustness (16%), while it performs well (up to 100%) for Llama-2-7B-Chat and Llama-3-8B-Instruct. However, given that ICA is not our main focus, these variations do not diminish the defense's effectiveness against the targeted attack types.

The MT-bench scores for DiffTextPure are slightly lower than other defenses, which is attributed to the length limitations of the current pretrained models [26]. This limitation affects performance on the benchmark but is expected to improve as future models handle longer contexts more effectively.

Overall, the results indicate that DiffTextPure can significantly enhance the resilience of large language models to various optimization-based adversarial attacks, offering a plug-and-play defense that maintains robustness across different model architectures and attack strategies.

## 5 Limitations

Although our defense provides certified lower bounds and outperforms previous baselines in white-box settings, it still faces several limitations that affect its efficiency and effectiveness in commercial, real-world black-box scenarios, as acknowledge as follows.

**Limitation 1: Defending against expertise-based attacks.** The core principle of our defense is to transform out-of-distribution data back into in-distribution data, and its certified guarantees are effective only when the length of the adversarial suffix is limited. However, expertise-based attacks, which utilize human-crafted prompts, often appear natural (i.e., have high likelihood) and are typically lengthy, rendering our theoretical guarantees less effective (see ICA in Table 1). This issue could potentially be addressed by integrating our defense with existing heuristic defenses.

**Limitation 2: Limited length.** In this paper, we utilize an off-the-shelf pretrained discrete diffusion model from [26]. However, due to constraints from its positional encoding, it only supports text with a length of less than 1024 tokens. To address this issue, we plan to adopt more advanced positional encoding methods, such as RoPE [40], and scale up the diffusion language models to further enhance their effectiveness.

**Limitation 3: High information-density data.** A critical limitation of our approach is that the forward process has some probability of destroying important information, particularly in cases of high information-density. This makes it challenging for diffusion models to fully recover the original content. For example, in mathematical problems, if key numerical values are altered during the forward process, their recovery becomes impossible since such values typically occur only once in the input text.

## 6 Conclusion

In this paper, we propose DiffTextPure, a novel defense mechanism that generalizes DiffPure to the discrete domain using discrete diffusion models. By applying both forward and reverse processes, DiffTextPure effectively mitigates adversarial attacks by transforming out-of-distribution inputs into in-distribution data, while preserving the utility of benign inputs. Our approach offers a plug-and-play solution with minimal computational overhead and a strong theoretical guarantee, making it highly practical for defending against optimization-based adversarial attacks. Experimental results confirm the efficiency and effectiveness of DiffTextPure in enhancing the security and robustness of LLM systems, paving the way for broader research and addressing the limitations.

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
