# OpenReview forum: "DiffTextPure: Defending Large Language Models with Diffusion Purifiers"
_NeurIPS.cc/2024/Workshop/SafeGenAi — SafeGenAi Poster_

### Official Review · Reviewer_tnrn · 2024-10-09
**Review of Submission32**

**Rating:** 6
**Confidence:** 3

**Review:**

Quality

Pros:
1. The paper presents an innovative approach that transferring a diffusion model, which is originally used in image to text tokens. This represents a creative use of diffusion models in the discrete domain.

2. The methodology is clearly described, including both forward and reverse diffusion processes, and empirical results demonstrate significant improvements in robustness against a range of adversarial attacks.

3. This paper provides a theoretical guarantee to certified robustness of their method.

Cons:

1. The computational overhead of performing discrete diffusion operations may limit the scalability of this method in large-scale real-world deployments, particularly for high-throughput applications.

2. DiffTextPure could not deal with ICA very well especially using Vicuna.


3. There is no study about trade-off between performance and safety. Since the DiffTextPure will change the input prompt, I assume it will hurt the performance.

Clarity

Pros:

1. The paper provides a clear introduction to the motivation for defending LLMs against adversarial attacks and the potential benefits of using diffusion purifiers.

Cons:

1. The authors may provide an example of ICA to show why the proposed method fails when meeting ICA and Vicuna.

2. Though the author claims that they have results for Llama-3, table 1 does not contain results for Llama-3.

Originality

Pros:

1. The application of discrete diffusion models for textual adversarial defense is novel and expands the applicability of diffusion models from the vision domain to natural language, which is an original contribution to the field.

2. By focusing on a plug-and-play approach that avoids joint training with LLMs, DiffTextPure provides a novel method that aims for broader applicability compared to other computationally expensive defense mechanisms.

Cons:

1. The idea of filtering adversarial prompts has parallels in other existing approaches that involve randomized smoothing or similar probabilistic transformations. The paper could have made a stronger case for how DiffTextPure significantly advances these methods beyond simple noise injection.

2. The idea of using diffusion models to defend against adversarial attacks has been proposed in the image domain. It seems the authors just combine discrete diffusion models with the idea of using diffusion models for defense.

Significance

Pros:

1. The ability to use DiffTextPure as an off-the-shelf defense without modifying the core LLM architecture is a practical contribution, potentially useful for real-world applications seeking lightweight security enhancements.

2. Experimental results show substantial improvements in robustness against several jailbreak techniques, suggesting that DiffTextPure could be a significant tool for improving the safety of language model deployments.

Cons:

1. The approach has limitations regarding expertise-based attacks, such as ICA. DiffTextPure shows reduced effectiveness against these attacks, which are common in real-world scenarios where adversaries use crafted, semantically coherent prompts. This weakens its overall practical significance.

2. There is no study about the trade-off between performance and safety. Using such a defense strategy will change the prompt directly and may destroy samples when a normal user wants to use in-context learning.

---

### Official Review · Reviewer_efRR · 2024-10-09
**Review of DiffTextPure**

**Rating:** 7
**Confidence:** 3

**Review:**

Inspired by the success of DiffPure in handling adversarial image attacks, the authors propose DiffTextPure, a novel approach that purifies adversarial textual inputs using a diffusion-based method. This approach works during a pre-processing phase, ensuring that adversarial perturbations in input text are removed before the LLM generates any output. The method is designed as a plug-and-play defense, meaning it can be applied without modifying or retraining the underlying large language models (LLMs). This makes it highly efficient in terms of computational overhead.

Strengths:
Efficiency and Applicability:
DiffTextPure operates in a pre-processing step, avoiding the need for joint training with the LLM or significant changes to its generation logic. This makes it a highly efficient and easy-to-use solution for improving adversarial robustness across various LLMs.
The plug-and-play nature allows broad applicability without increasing the computational costs, making it ideal for practical, large-scale deployment.
Strong Experimental Results:
The method achieves these results with minimal computational cost, proving its practicality.
Theoretical Guarantees:
The authors provide certified robustness guarantees, offering a theoretical foundation for the defense method. These guarantees increase the credibility of DiffTextPure in practical applications that require formal safety measures.

Weaknesses:
Limited Effectiveness Against Expertise-Based Attacks:
The method performs poorly against expertise-based attacks, where human-crafted prompts appear natural and semantically complex. These attacks are difficult to defend against because they often evade detection by mimicking benign inputs closely.
Token Length Limitation:
DiffTextPure is constrained by the pre-trained discrete diffusion model, which limits it to processing texts of fewer than 1024 tokens. This limitation reduces its effectiveness in scenarios where long text inputs are common.
Loss of Critical Information in High Information-Density Data:
In cases involving high information-density inputs (e.g., mathematical texts or data-heavy inputs), the forward diffusion process can destroy key information that may not be fully recovered in the reverse process. This issue is especially problematic for inputs where certain elements (like numerical values) occur only once.


Suggestions for Improvement:
Analysis of Non-Optimization-Based Attacks:
A deeper analysis is needed to explain why DiffTextPure performs poorly against attacks like In-Context Attacks (ICA). This additional analysis would help clarify the method's limitations and potentially inspire improvements to handle these more effectively.
Coherence Between Diffusion Explanation and DiffTextPure Introduction:
The description of diffusion models and the subsequent introduction of DiffTextPure should be more cohesive. Currently, these sections feel somewhat disjointed, making the flow of the paper less smooth. A more structured transition between these two parts would improve readability and help the reader better understand the core concepts.

---

### Official Review · Reviewer_JuoM · 2024-10-09

**Rating:** 7
**Confidence:** 3

**Review:**

This paper addresses the problem of defending LLMs against jailbreaking attacks, i.e., adversarial suffixes attached to the input to trick them into generating harmful or unwanted content. The authors claim that the existing defense mechanisms face practical limitations, since they need to modify the generation logic or significantly increase the generation cost. In this paper, inspired by the success of diffusion modules for defending against vision adversarial examples, the authors develop a plug-and-play diffusion purification defense, DiffTextPure, specialized for defending against textual jailbreaking attacks.

The paper is well-written, clear, and detailed. The authors propose a comprehensive related work and a clear methodology. They evaluate their approach on a large set of different attacks and show that DiffTextPure improves over the state of the art.